# Wearable Intelligent Machine Learning Rehabilitation Assessment for Stroke Patients Compared with Clinician Assessment

**DOI:** 10.3390/jcm11247467

**Published:** 2022-12-16

**Authors:** Liquan Guo, Bochao Zhang, Jiping Wang, Qunqiang Wu, Xinming Li, Linfu Zhou, Daxi Xiong

**Affiliations:** 1School of Biomedical Engineering (Suzhou), Division of Life Sciences and Medicine, University of Science and Technology of China, Hefei 230052, China; 2Suzhou Institute of Biomedical Engineering and Technology, Chinese Academy of Sciences, Suzhou 215163, China; 3Department of Rehabilitation Medicine, Tangdu Hospital Airforce Medicine University, Xi’an 710032, China; 4Department of Rehabilitation Medicine, Xi’an Gaoxin Hospital, Xi’an 710065, China; 5Department of Respiratory and Critical Care Medicine, The First Affiliated Hospital, Nanjing Medical University, Nanjing 210029, China

**Keywords:** wearable devices, intelligent rehabilitation, rehabilitation assessment, Fugl–Meyer, stroke

## Abstract

In order to solve the shortcomings of the current clinical scale assessment for stroke patients, such as excessive time consumption, strong subjectivity, and coarse grading, this study designed an intelligent rehabilitation assessment system based on wearable devices and a machine learning algorithm and explored the effectiveness of the system in assessing patients’ rehabilitation outcomes. The accuracy and effectiveness of the intelligent rehabilitation assessment system were verified by comparing the consistency and time between the designed intelligent rehabilitation assessment system scores and the clinical Fugl–Meyer assessment (FMA) scores. A total of 120 stroke patients from two hospitals participated as volunteers in the trial study, and statistical analyses of the two assessment methods were performed. The results showed that the R^2^ of the total score regression analysis for both methods was 0.9667, 95% CI 0.92–0.98, *p* < 0.001, and the mean of the deviation was 0.30, 95% CI 0.57–1.17. The percentages of deviations/relative deviations falling within the mean ± 1.96 SD of deviations/relative deviations were 92.50% and 95.83%, respectively. The mean time for system assessment was 35.00% less than that for clinician assessment, *p* < 0.05. Therefore, wearable intelligent machine learning rehabilitation assessment has a strong and significant correlation with clinician assessment, and the time spent is significantly reduced, which provides an accurate, objective, and effective solution for clinical rehabilitation assessment and remote rehabilitation without the presence of physicians.

## 1. Introduction

Stroke is a disease in which cerebral blood circulation is impaired, and brain tissue function and structure are damaged due to the obstruction or rupture of cerebral blood vessels. It is the third leading cause of death and the second leading cause of disability worldwide, and its high morbidity, mortality, and disability rates bring economic burden and mental stress to society and families [1]. According to the China Stroke Prevention and Treatment Report 2021, in 2020, the standardized prevalence rate of stroke in people over 40 years old in China was 2.61%, and the incidence rate was 505.23/per 100,000. There were about 17.8 million stroke patients and 3.4 million new stroke patients each year, compared with about 13.7 million in the world.

According to statistics, about 70–85% of first-time stroke patients have limb movement dysfunction, which seriously affects quality of life and burdens both society and the family of the patient heavily. Timely and effective rehabilitation training can help the patient to recover certain motor functions [2].

Rehabilitation assessment is an essential part of rehabilitation treatment, which is the judgment of patients’ functional status and potential ability and is also the process of data collection, quantification, analysis, and comparison with the usual standards for all aspects of patients. Based on the results of the rehabilitation assessment, the best rehabilitation treatment plan is formulated, and the rehabilitation assessment is conducted periodically at regular intervals to assess the effect of rehabilitation and improve the treatment plan.

Currently, the clinical assessment of motor function in patients with stroke often uses scale assessment methods, such as the Fugl–Meyer assessment (FMA) scale [3,4] and the Brunnstrom assessment method [5,6]. These scale assessment methods, which rely on the physician’s examination and observation, are widely used in clinical practice but still suffer from several problems. First, the assessment results are easily influenced by subjective factors of the rehabilitation physicians. The assessment results of different assessors at different times may vary, and the accuracy of the rehabilitation assessment often depends on the experience and level of the rehabilitation physician. Second, the assessment items are numerous and time-consuming. They require the real-time participation of the rehabilitation physician, and a complete rehabilitation assessment takes 30–45 min, which is often difficult to achieve with relatively scarce rehabilitation resources and home-based or community-based rehabilitation [7]. Third, the scale grading is coarse, with a “ceiling effect” (patients progress after a particular stage of rehabilitation, but the scale score remains mainly unchanged) [8]. As a result, the assessment results do not accurately reflect the process and details of limb movement rehabilitation, which is not conducive to the generation of individualized and refined rehabilitation treatment programs.

To solve those problems, more and more researchers have developed applications based on new technologies to assess rehabilitation results and combined these with machine learning or artificial intelligence algorithms to realize movement recognition [9,10,11], movement classification [12,13,14], and movement assessment [15,16,17]. The main research is divided into methods based on wearable devices [18,19,20] and visual devices such as Kinect [21,22,23]. Among them, the method based on wearable devices has become a hot spot and trend in rehabilitation assessment research due to higher accuracy, stronger anti-interference ability, low cost, and convenient use.

Wang et al. [24] showed that the collection of patient kinematic data and surface electromyography (EMG) signals by wearable sensors and the collaborative activation characteristics of motor injury were analyzed. A multi-mode fusion scheme was designed to analyze the functional score of upper limb movement, and the accuracy of movement classification was improved to 96.06%. However, the test was difficult in meeting the measurement and assessment of stroke patients with severe disabilities. Knorr et al. [25] designed a linear and nonlinear feature to assess subjects with upper limb motor dysfunction after stroke. The severity of functional defects and motor injuries was determined by quantifying the accelerometer signals of the arms and hands. However, the study only analyzed two upper limb functional movements, and there was a lack of assessment of the motor function of the patients’ whole body. Held et al. [26] completed the clinical gait assessment of patients through the combination of intelligent glasses and a sensor-based motion capture system. However, the research showed that the calibration technology and direction drift problem limited the clinical assessment and application of the system.

The above research is only a preliminary exploration of the quantitative assessment of the motor function of stroke, and a series of problems have not been further studied, such as how to realize the quantitative assessment of motor function, and what the mapping relationship is with the commonly used clinical rehabilitation assessment scale.

According to the above problems, a wearable motion capture system based on nine-axis inertial sensor Inertial Measurement Units (IMUs) and Flex sensors was designed, which was worn on the affected limbs, and the patient’s rehabilitation movement data were collected and recorded in real-time wirelessly. Through multi-sensor information fusion, feature extraction, and a machine learning algorithm, a rehabilitation assessment model consistent with the clinical Fugl–Meyer Assessment (FMA) score was established, and the quantitative rehabilitation score was given. Finally, the accuracy and effectiveness of the intelligent rehabilitation assessment system were verified by comparing the consistency and time of the designed intelligent rehabilitation assessment system and the rehabilitation physician’s rehabilitation assessment of stroke through clinical trials.

## 2. Methods

### 2.1. Participants and Protocol

#### 2.1.1. Participants

For this clinical study, stroke patients with limb motor dysfunction 15–180 days after onset were selected as subjects for the study at Tangdu Hospital and Xi’an Gaoxin Hospital and were approved by the Ethics Committees of Tangdu Hospital and Xi’an Gaoxin Hospital. In addition, all patients participating in the study had signed an informed consent form. Clinical trial registration number: ChiCTR2200061310, website: https://www.chictr.org.cn/showprojen.aspx?proj=171734 (accessed on 20 November 2022).

Patient inclusion criteria included: (1) stroke patients diagnosed by CT or MRI within 90 days; (2) aged between 30–75 years, male or female; (3) limb motor dysfunction caused 15–180 days after stroke onset (recovery period); (4) recovered patients with a Brunnstrom upper or lower limb classification of stage II-VI with stable disease; (5) clear cognition and able to follow the study protocol; (6) subjects who understood the purpose of the study, showed sufficient compliance with the study protocol, and signed an informed consent form.

The following patients were excluded: (1) those with obvious cognitive and conscious disorders that could not complete the Fugl–Meyer assessment; (2) those with other major limb injuries, such as fractures, severe arthritis, amputations, etc.; (3) formation of limb joint contracture; (4) patients with disabilities prescribed by law (such as blindness, deafness, dumbness, intellectual disability, mental disorder, physical disability); (5) cases complicated with severe primary diseases such as cardiovascular, liver, kidney, and hematopoietic systems, mentally ill patients, and other circumstances that the researcher considered inappropriate for participating in the experiment.

#### 2.1.2. Procedure

The patients who met the test standards were scored by the intelligent assessment system and physicians, respectively, and the assessment scores and usage time of the system and physicians were recorded, respectively. In order to study the rehabilitation assessment without the participation of physicians, during the whole rehabilitation assessment process, a rehabilitation physician or therapist was by the patient’s side, only to ensure their safety and not to participate in the rehabilitation assessment process of the patients.

### 2.2. Intelligent Rehabilitation Assessment System

The block diagram of the intelligent rehabilitation assessment system based on wearable devices and machine learning is shown in Figure 1. Figure 1A–C are the schematic diagrams of upper limbs, hands, and lower limbs with wearable devices on the affected side, respectively. Figure 1D is the schematic diagram of the rehabilitation assessment process. Stroke patients wore wearable devices according to the requirements of the instructions and completed the corresponding rehabilitation assessment movements according to the standard movement video in the rehabilitation assessment software. The wearable sensors collected the patients’ movement data, received it through a Zigbee wireless receiver, and transmitted it to the computer through USB. The PC-side rehabilitation assessment software collected, stored, and displayed the data. The signal quality was improved by pre-processing such as sliding filtering, and then the movement characteristics of patients after multi-sensor fusion were extracted. The extracted multiple movement features were matched with the standard template data by Dynamic Time Warping (DTW) to obtain a single movement score. Then, all the single movement scores were used as model input, and the rehabilitation assessment model was established by machine learning algorithms with the rehabilitation physician’s FMA scores as the label. For new rehabilitation assessment movement data, the established rehabilitation assessment model can directly obtain the quantitative rehabilitation score that is consistent with the clinical rehabilitation physician’s FMA score.

Wearable equipment mainly included a nine-axis inertial sensor module for upper- and lower-limb rehabilitation assessment, rehabilitation gloves for hand rehabilitation assessment, the Zigbee wireless receiver, etc. (Xi’an Libang Contmedu Medical Technology Co., Ltd., Xi’an, China, model: KMD-T001). The wearable devices were tested by third-party inspectors and met clinical research requirements in terms of signal quality, EMC, and safety. In terms of measurement accuracy, according to the third-party inspection report, the measurement accuracy of angle, acceleration, and angular velocity were all above 95%. Each IMU module contains nine-axis motion sensors, including a three-axis accelerometer, a three-axis angular velocity meter, and a three-axis magnetometer. In the rehabilitation assessment, four IMU modules were fixed on the upper arm, forearm, thigh, and calf. Rehabilitation gloves contained one IMU and five Flex sensors to monitor the movement of wrists and individual fingers. The glove was designed with a left hand and a right hand, in three sizes of large, medium, and small, for use by different patients.

The battery of the wearable device is 400 mAh, and the power consumption is 20 mAh. The battery can be charged for 20 h continuously, which can meet the needs of rehabilitation institutions for more than two days and home patients for more than ten days. Each sensor is networked through the Zigbee2007 wireless communication protocol, which is very convenient for expansion and synchronous data acquisition. The sampling rate for each wearable device is 30 times per second, which was sufficient for data collection, analysis, and rehabilitation assessment.

### 2.3. Machine Learning Model

#### 2.3.1. Rehabilitation Assessment Movements

Considering that complete assessment using the FMA scale is a time-consuming task, in the study of intelligent rehabilitation assessment, manual selection or machine learning algorithms are often used to select the most relevant items to replace the complete assessment items [27]. In the present study, patients with Brunnstrom stage II and above were selected under the guidance of clinicians, and all patients could perform nerve reflex activity effectively. This study referred to the items in the FMA scale and selected and designed 11 categories according to the characteristics of patients and task similarity. A total of 32 movements were used as assessment tasks, including six joint movements of the upper-limb flexor, three joint movements of the upper-limb extensor, three joint movements of the upper limb, three upper-limb disengagement collaborative movements, five wrist stability movements, seven finger function movements, one upper-limb coordination and speed movement, three lower-limb flexor joint movements, four lower-limb extensor joint movements, two lower-limb joint movements, two lower-limb disengagement collaborative movements, and one lower-limb coordination and speed movement.

The 32 movements selected above were used to model the mapping relationship between the 32 movements and the clinical FMA score based on the wearable device and machine learning algorithms, thus giving assessment results consistent with the FMA score.

#### 2.3.2. Establishment of Machine Learning Rehabilitation Assessment Model

The human movement sequence is a time series, and time series similarity comparison is a common time series analysis method. However, the Euclidean distance on the time axis cannot be used directly for the similarity measure of time series [28]. Therefore, this study used the Dynamic Time Warping (DTW) algorithm to calculate the similarity of patients’ movement data with the standard template data.

The DTW algorithm provided a method to measure the similarity of two unequal-length sequences by stretching and compressing the time axis to remove temporal distortions, thus finding an optimal matching path and minimalizing the cumulative matching distance between the two sequences. Using the distance of normalized paths to assess the similarity between sequences can overcome the problem of unmatched sequences due to different signal lengths [29]. It was known from clinical experience that the rehabilitation movements of stroke patients vary in speed due to limb movement disorders, resulting in inconsistent signal lengths and problems with offsetting movement trends. Therefore, DTW was well-suited for the recognition of rehabilitation movements. A brief description of the DTW algorithm is given below.

Two time series with lengths *M* and *N* are given:(1)$$X=\left[x_{1},x_{2},...,x_{M}\right]$$
(2)$$Y=\left[y_{1},y_{2},...,y_{N}\right]$$

The lengths of *X* and *Y* can be unequal, but the elements in each sequence must have the same dimension. For example, suppose sequence *X* is a reference sequence and sequence *Y* is a test sequence.

The matching path $W=[w_{1},w_{2},...,w_{L}]$ is used to describe the matching relationship between the elements of sequence *X* and *Y*, $w_{l}=(i_{l},j_{l})$, indicating that the elements in sequence *X* match the elements in sequence *Y*. The number of rows of the matching path *W* satisfies max (*M*,*N*) ≤ L < *M* + *N* − 1, and the set of all matching paths between sequence *X* and *Y* is defined as $W_{M,N}$.

The matching distance $d_{W}(X,Y)$ is the cumulative distance of sequence *X* and *Y* under the matching path *W*, that is, the sum of the distances of all matching elements:(3)$$d_{W}\left(X,Y\right)=\sum_{l=1}^{L}d\left(x_{i_{l}},y_{i_{l}}\right)$$

In general, the square of the Euclidean distance is used to represent the distance between elements, which is $d(x_{i_{l}},y_{i_{l}})={\left(x_{i}-y_{i}\right)}^{2}$.

The DTW distance is the minimum matching distance between sequences *X* and *Y* under the satisfaction of the matching path constraint, that is:(4)$$DTW\left(X,Y\right)=\sqrt{d_{W}\cdot \left(X,Y\right)}=\min\left\{\sqrt{d_{W}\left(X,Y\right)}\left|\left.W\in W_{M,N}\right\}\right.\right.$$

The matching path *W** corresponding to the minimum matching distance is the optimal matching path. We first calculated the cumulative matching distance matrix between two sequences in this study. Then, according to the monotonic continuity constraint in path matching, the cumulative distance of the matching path is:(5)$$D_{i,j}=d\left(x_{i},y_{i}\right)+\min(D_{i-1,j-1},D_{i-1,j},D_{i,j-1})$$

According to the definition of the DTW distance and the cumulative matching distance, the DTW distance of sequence *X* and *Y* is:(6)$$DTW\left(X,Y\right)=D_{M,N}$$

The DTW distance is a non-negative number whose smaller value indicates greater similarity between two sequences, and vice versa.

The DTW distance is converted to the range [0,1] according to the linearization method with the following normalization formula:(7)$$X=\frac{X_{i}-X_{\min}}{X_{\max}-X_{\min}}$$
where *X*_max_ is the maximum value of the sample data, *X*_min_ is the minimum value of the sample data, *X_i_* is the raw data, and *X* is the normalized data. For the rehabilitation movement assessment, *X*_max_ is the DTW distance when there is no movement at all, and *X*_min_ is the DTW distance of the rehabilitation physician’s standard rehabilitation movement, which is 0. Therefore, the rehabilitation movement score can be obtained by the following formula and standardized to 0–100 points:(8)$$Score=(\sum_{i=1}^{m}\frac{X_{i}}{X_{\max}}*\delta_{i})*100\;$$
where $\delta_{i}$ is the weight occupied by a certain feature, such as elbow joint angle in the rehabilitation movement, which is obtained after information entropy calculation and normalization.

Considering that the single movement score of the intelligent rehabilitation assessment system is 0–100, whereas the corresponding FMA score has only three values of 0, 1, and 2, direct correspondence will lead to a large error. Therefore, the scores of 32 movements were used as the input features of the machine learning algorithm to obtain the total system score, and this were compared with the rehabilitation physician’s FMA score to evaluate the designed intelligent rehabilitation assessment system.

The machine learning rehabilitation assessment model building process is shown in Figure 2. First, the stroke patient completed 32 rehabilitation assessment movements with the wearable devices according to the standard video. Then, after preprocessing by filtering and effective movement extraction, the effective movement cycle signal was matched with the standard template to obtain the DTW distance, and all the DTW distances and corresponding weights of multiple information, such as angle, angular velocity, and acceleration, were calculated to obtain the movement score. Further, the 32 movement scores were used as the input features, and the rehabilitation physician’s FMA scores were used as the labels to build a mapping model with the clinical FMA score results by the support vector regression (SVR) machine learning algorithm. Finally, for new patients, the established machine learning model was used to obtain assessment results that were consistent with the physician’s FMA score.

### 2.4. Rehabilitation Assessment Software

The rehabilitation assessment system software based on wearable devices and machine learning is shown in Figure 3. Patients wore wearable devices and completed the corresponding movements according to the standard movement video on the software interface. The next item was performed after completing the single movements, until all 32 rehabilitation assessment tasks were completed. After completing all the projects, the system automatically gave the rehabilitation assessment score according to the established machine learning model and generated the assessment report offline. Furthermore, all data and results were stored and uploaded to the remote server and cloud rehabilitation platform for physicians to view, analyze and guide the remote patients’ recovery. In the assessment of motor function, the joint angles and the acceleration and angular velocity of the limbs are the key information, which can be directly obtained or obtained by simple calculation from the IMU or Flex sensor, with an accuracy of more than 95%. In addition, considering the time-consuming 3D reconstruction and kinematic analysis and the high requirements for the computer even with GPU, 3D kinematic analysis was not performed in the rehabilitation assessment software.

### 2.5. Data Analysis

In the two rehabilitation medical centers, 120 subjects were actually enrolled. Among these, 60 cases were enrolled in Tangdu Hospital and 60 in Xi’an Gaoxin Hospital. For all 120 stroke patients, motor function was assessed by the intelligent rehabilitation assessment system and the rehabilitation physician, respectively. The intelligent rehabilitation assessment system used the designed wearable device-based and machine learning model for scoring, and the rehabilitation physician assessed according to the FMA scale.

SAS software (version 9.4) was used for analysis in this study. After all rehabilitation assessments were completed, regression analysis was performed on the intelligent rehabilitation assessment system score and physician’s FMA score to calculate the intercept, slope, 95.0% CI of the slope, and R^2^ value. The regression diagram was drawn with the rehabilitation physician’s score as the *x*-axis and the intelligent rehabilitation assessment system’s score as the *y*-axis; calculation of the intelligent rehabilitation assessment system score’s deviation, calculation of the mean and 95% confidence interval, drawing Bland-Altman diagram. In addition, the time required to record the score was counted, and a student’s *t* test was used to compare the difference between the time taken by the physician and the time taken by the intelligent rehabilitation assessment system.

## 3. Result

### 3.1. Clinical Characteristics of Patients

In this study, 369 subjects were assessed for eligibility, and 246 subjects were excluded, of which 231 did not meet the inclusion criteria, 15 were unwilling to participate, and 3 had other reasons. Finally, 120 stroke patients were recruited as subjects, including 86 males and 34 females. Figure 4 shows the study flow of the enrollment. As shown in Table 1, the average age of patients was 56.47 ± 9.70, height 167.80 ± 6.95 cm, weight 69.68 ± 10.95 kg, pulse 77.59 ± 7.95, systolic pressure 128.8 ± 12.71, diastolic pressure 81.75 ± 9.25, and respiratory rate 18.89 ± 1.68.

### 3.2. Analysis of System Assessment Results

All 120 patients completed the clinic trial according to their rehabilitation assessment. The comparison results of the intelligent rehabilitation assessment system score and the physician’s Fugl–Meyer score are shown in Table 2. The average total score of the intelligent rehabilitation assessment system was 41.78, the 95% CI was 37.19–46.38, and the average total score of the physician’s FMA was 41.48, and the 95% CI was 36.72–46.25. The average upper-limb score of the intelligent rehabilitation assessment system was 24.62, the 95% CI was 21.27–27.97, and the average upper limb score of the physician’s FMA was 24.03, with the 95% CI being 20.55–27.51. The average of the lower-limb score of the intelligent rehabilitation assessment system was 17.17, the 95% CI was 15.50–18.83, and the average of the FMA lower-limb score of the physician’s FMA was 17.45, with the 95% CI being 15.86–19.04.

The regression analysis results of the intelligent rehabilitation assessment system score and the physician’s FMA score are shown in Table 3 and Figure 5. The regression analysis of the total score of the intelligent rehabilitation assessment system and the physician’s FMA showed that the intercept of the fitting curve was 2.43, the slope was 0.95, the 95.0% CI of the slope was 0.92–0.98, and the R^2^ was 0.9667, *p* < 0.001; the regression analysis of the upper-limb score of the intelligent rehabilitation assessment system and the physician’s FMA showed that the intercept of the fitting curve was 1.98, the slope was 0.94, the 95% CI of the slope was 0.90–0.98, R^2^ was 0.9577, *p* < 0.001; the regression analysis of the lower-limb score of the intelligent rehabilitation assessment system and the physician’s FMA showed that the intercept of the fitting curve was 0.22, the slope was 0.97, the 95.0% CI of the slope was 0.90–1.04, and the R^2^ was 0.8575, *p* < 0.001.

In the regression analysis of the intelligent rehabilitation assessment system’s score and the physician’s FMA score, all the *p* values of the total score, the upper-limb score, and the lower-limb score regression curve were less than 0.001, indicating that the system score was significantly correlated with the physician’s FMA score. In addition, the R^2^ of the total score was the highest, and the system had the best fitting effect on the assessment of the whole-body limb motor function of patients, and the R^2^ of the lower limbs was the lowest. The reason may be that the lower-limb items in the FMA project were relatively few (33 upper limbs and 17 lower limbs).

### 3.3. Analysis of System Deviation and Relative Deviation

In this study, the deviation between the score of the intelligent rehabilitation assessment system and the score of the physician’s Fugl–Meyer was analyzed, as shown in Table 4 and Figure 6 and Figure 7. The mean deviation between the total score of the intelligent rehabilitation assessment system and the total score of the physician’s FMA was 0.30, the 95% CI was −0.57–1.17, the relative deviation (%) was 3.49, the 95% CI was 0.50–6.48; the mean deviation between the upper-limb score of the intelligent rehabilitation assessment system and the FMA upper-limb score of the physician was 0.58, the 95% CI was −0.13–1.30, the relative deviation (%) was 7.20, the 95% CI was 2.36–12.04; the mean deviation between the lower-limb score of the intelligent rehabilitation assessment system and the FMA lower limb score of the physician was −0.28, and the 95% CI was −0.91–0.35; the relative deviation (%) was −0.78, and the 95% CI was −5.44–3.88.

In the analysis of the deviation and relative deviation of the assessment effect of the system, the absolute value of the deviation and relative deviation of the lower limb was the smallest. In other words, the dispersion degree of the lower-limb score was the smallest, and the deviation degree from the physician’s score was the smallest. The system showed good consistency with the physician’s FMA score, because the lower limb movement was close to daily life activities, and both the physician’s score and the system score could accurately assess the movement of patients. In addition, the absolute value of upper-limb movement deviation and relative deviation was the largest, because there were multiple movement degrees of freedom in upper-limb movement, and some serious patients had a certain compensation in performing tasks, although they had been required to remain upright. In summary, the relative deviation between the rehabilitation score of the system and the FMA score of the rehabilitation physician was less than or equal to 7.20%, which was within the acceptable range.

### 3.4. Data Distribution Analysis of System Deviationand Relative Deviation

In this study, the distribution of deviations and relative deviations of the intelligent rehabilitation assessment system’s score and the physician’s FMA score were analyzed, as shown in Table 5. The percentage of deviation and relative deviation between the total score of the intelligent rehabilitation assessment system and the physician’s FMA scale within the mean ± 1.96 SD (95% consistency limit) of deviation and the relative deviation was 92.50% and 95.83 %, respectively. The percentage of deviation and relative deviation between the upper-limb score of the intelligent rehabilitation assessment system and the physician’s FMA score within the mean ± 1.96 SD (95% consistency limit) of deviation and the relative deviation was 95.00% and 95.83%, respectively. The percentage of deviation and relative deviation between the intelligent rehabilitation assessment system’s lower-limb score and the physician’s FMA score within the mean ±1.96 SD (95% consistency limit) of deviation and the relative deviation was 95.83% and 96.67%, respectively.

In the analysis of the distribution of score deviations and relative deviations of the intelligent rehabilitation assessment system, the number of patients with lower-limb score deviation and relative deviation outside the limits of agreement was the lowest, reaching more than 95% of the clinically acceptable range. In contrast, the number of patients with total score deviation and relative deviation outside the consistency limits was the highest, which was because the total score included both upper- and lower-limb items, and the total score deviation and relative deviation increased, resulting in a relatively high distribution of patients outside the limits. However, the overall number of patients within the consistency limits reached more than 90%, indicating that the intelligent rehabilitation assessment system can be effectively used for clinical and remote rehabilitation assessment.

### 3.5. Analysis of Time Used for Scoring and Safety Report

The time required for the intelligent rehabilitation assessment system’s score and the physician’s FMA score were compared and analyzed by *t*-test. The statistical results showed that the time for the intelligent rehabilitation assessment system’s score was 13.82 ± 6.61 (min), and the 95% CI was 12.62–15.01, and the time for the physician’s FMA score was 21.26 ± 7.79 (min), with the 95% CI being 19.85–22.67 (min). Considering that the preparation time required for using the intelligent rehabilitation assessment system was about 1 min, including putting on the devices and opening software, 1 min preparation time was included in the results and reports. The difference between the two groups was statistically significant (*p* < 0.05), indicating that the intelligent rehabilitation assessment system could significantly shorten the rehabilitation assessment time of patients.

In addition, in terms of safety, no adverse events related to the intelligent rehabilitation assessment system occurred throughout the clinical study. Therefore, the designed intelligent rehabilitation assessment system based on wearable devices and machine learning had good safety when used for rehabilitation assessments of patients with limb motor dysfunction such as stroke.

## 4. Discussion

In this study, we designed an intelligent rehabilitation assessment system to accurately and quickly assess the rehabilitation of patients with limb motor dysfunction such as stroke. The system is based on multiple wireless wearable devices of a nine-axis IMU and a Flex sensor, and the rehabilitation movement data of patients were collected by Zigb ee wireless networking technology. The DTW algorithm and machine learning algorithm were adopted to establish an intelligent rehabilitation assessment model consistent with the clinical FMA score by completing 32 movement tasks. The results of 120 clinical trials showed that the designed intelligent rehabilitation assessment system based on wearable devices and machine learning could accurately, objectively, and rapidly realize the rehabilitation assessment and effectively solve the defects of excessive time consumption, strong subjectivity, and coarse classification of the clinical scale rehabilitation assessment.

Compared with most rehabilitation assessment techniques, such as an upper-limb motor function assessment system based on a desktop rehabilitation robot [30], a wrist movement assessment system based on force feedback [31], and a stroke daily life assessment system based on a textile platform [32], the wearable intelligent rehabilitation assessment system designed in this study could give quantitative rehabilitation assessment results and generate analysis reports, which could be used not only for physicians to accurately, objectively, and quickly assess the rehabilitation status of patients but also for patients’ immediate incentive feedback. Multi-source wearable sensors provided rich data to support the overall system architecture and intelligent assessment function. Based on satisfying different environments and postures, wearable devices, such as those for standing, sitting, and lying, achieved complete patient motor function assessment.

This study adopted the integrated learning model based on DTW and SVR in assessment algorithms. The DTW algorithm calculated the similarity between each rehabilitation assessment movement sequence and the corresponding standard data template sequence. Compared with the recognition algorithm based on probability statistics, the recognition algorithm based on the DTW model had a simple structure and a simple training process. It could assess the quality of rehabilitation assessment tasks from deep characteristics. The SVR machine learning algorithm was robust to outliers, with excellent generalization ability and high prediction accuracy. The experimental results showed that the designed intelligent rehabilitation assessment model based on ensemble learning had a high consistency with the FMA in limb motor function rehabilitation assessment. 

In this study, the subsequent signal processing was very sensitive to the initial position of wearable sensors, and incorrect wear may lead to a deviation in rehabilitation assessment results. For more serious patients, body compensation may affect the accuracy of the assessment, and future research will eliminate the deviation caused by movement compensation by adding body sensors. In addition, we will study more advanced artificial intelligence algorithms, such as deep learning algorithms, to further improve the accuracy and refinement of the intelligent rehabilitation assessment system.

The driving force of this study was to establish an intelligent rehabilitation assessment model and system based on wearable devices and machine learning for patients with limb motor dysfunction such as stroke. The system can be used as a clinical rehabilitation assessment and remote home rehabilitation assessment tool, which can effectively solve the shortcomings of the current clinical assessment scale, such as excessive time consuming, strong subjectivity, and coarse classification. In the future, the system will realize more exercise scales and rehabilitation assessment projects and carry out experiments and clinical research on a wider range of people.

## 5. Conclusions

In this study, we found that the designed intelligence rehabilitation assessment system based on wearable devices and machine learning was highly consistent with physician’s FMA scores in terms of the rehabilitation assessment. It took significantly less time than physicians and had good safety parameters. Therefore, the intelligent rehabilitation assessment system could effectively help physicians and patients conduct an accurate, objective, and rapid rehabilitation assessment to save valuable rehabilitation medical resources. In the future, it will be approved by the CFDA and used in clinical rehabilitation assessment, such in rehabilitation hospitals and other rehabilitation institutions, as well as in remote home environments.

## Figures and Tables

**Figure 1 jcm-11-07467-f001:**
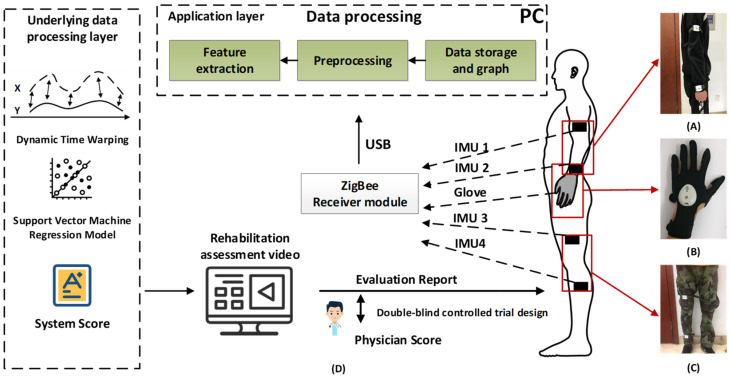
Rehabilitation assessment process based on wearable devices and machine learning. Upper limb-wearing instructions (**A**), glove-wearing instructions (**B**), and lower limb-wearing instructions (**C**) are the schematic diagrams of upper limbs, hands, and lower limbs with wearable devices on the affected side, respectively. Rehabilitation assessment process (**D**) is the schematic diagram of the rehabilitation assessment process.

**Figure 2 jcm-11-07467-f002:**
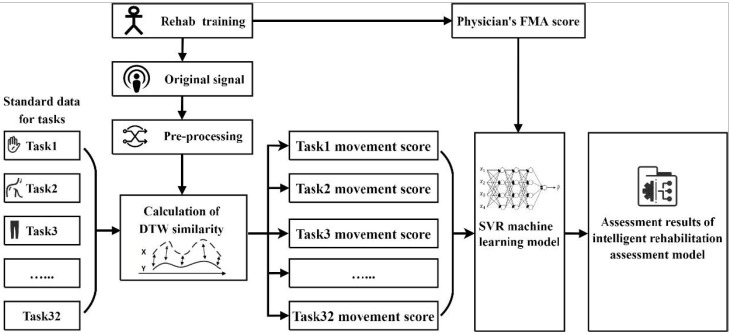
Establishment process of machine learning rehabilitation assessment model.

**Figure 3 jcm-11-07467-f003:**
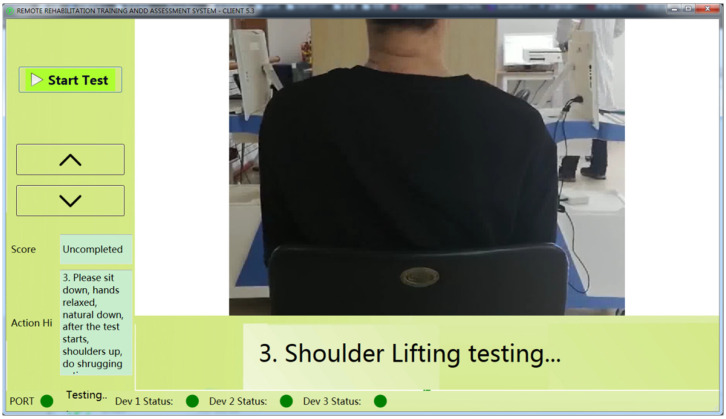
Rehabilitation assessment client software.

**Figure 4 jcm-11-07467-f004:**
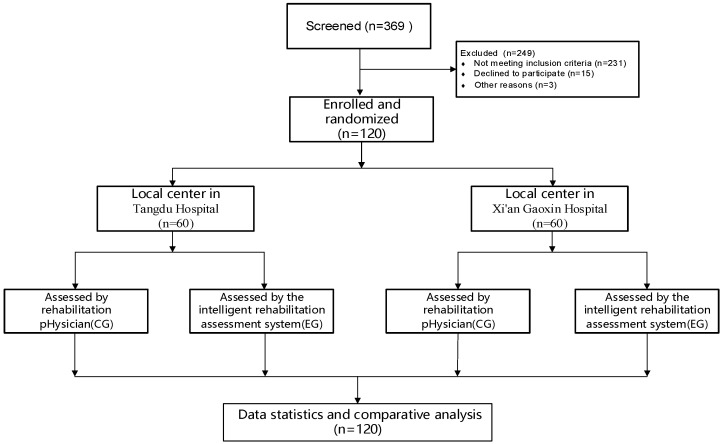
Flow chart of experiment in this study.

**Figure 5 jcm-11-07467-f005:**
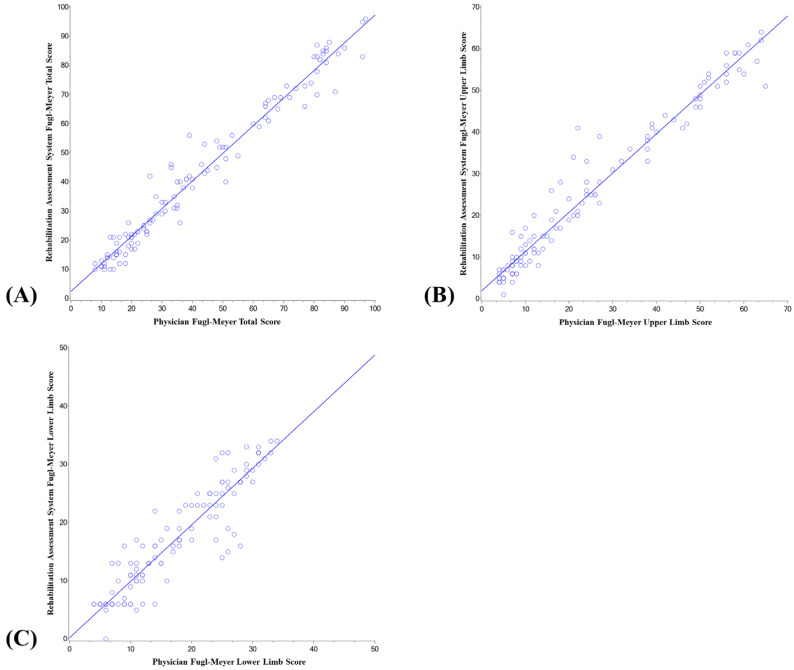
Intelligent rehabilitation system score and physician’s FMA score regression analysis chart. Total score (**A**), upper-limb score (**B**), and lower-limb score (**C**) are the regression analysis results of the intelligent rehabilitation assessment system score and the physician’s FMA score.

**Figure 6 jcm-11-07467-f006:**
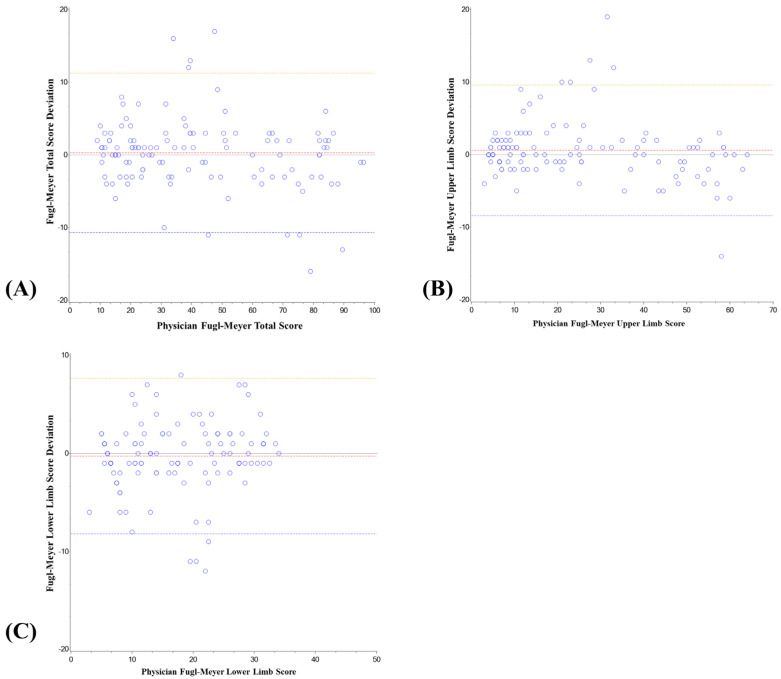
Intelligent rehabilitation system score and physician’s FMA score deviation analysis chart. Total score (**A**), upper−limb score (**B**), and lower−limb score (**C**) are the results of deviation between intelligent rehabilitation system score and physician’s FMA score.

**Figure 7 jcm-11-07467-f007:**
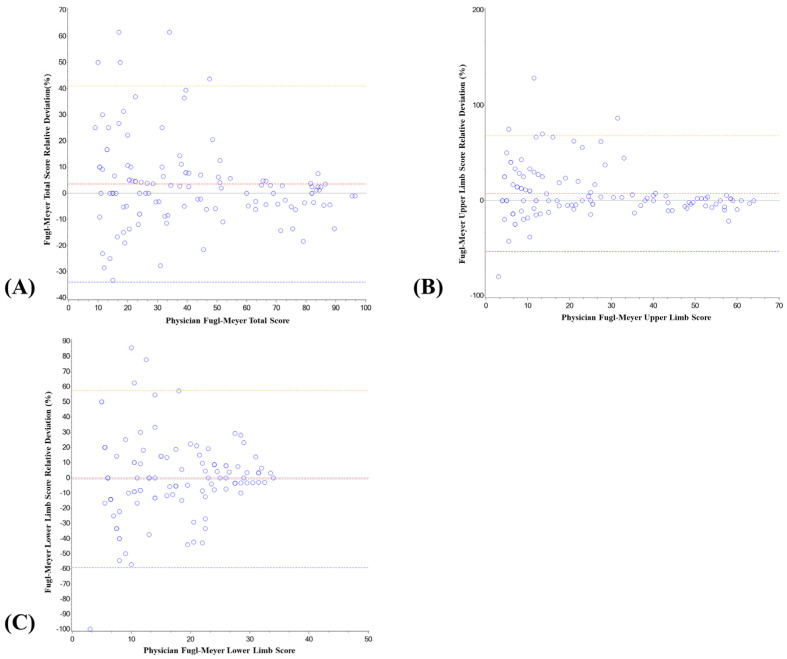
Intelligent rehabilitation system score and physician’s FMA score relative deviation analysis chart. Total score (**A**), upper−limb score (**B**), and lower−limb score (**C**) are the results of relative deviation between intelligent rehabilitation system score and physician’s FMA score.

**Table 1 jcm-11-07467-t001:** Statistics of baseline characteristics of subjects.

Characteristics	Mean (SD)	Min–Max	95% CI
Age	56.47 ± 9.70	33.00–74.00	54.71–58.22
Height	167.80 ± 6.95	150.00–180.00	166.50–169.00
Weight	69.68 ± 10.95	48.00–114.00	67.69–71.66
Pulse	77.59 ± 7.95	60.00–110.00	76.16–79.03
Systolic pressure	128.8 ± 12.71	92.00–116.00	126.50–131.10
Diastolic pressure	81.75 ± 9.25	57.00–107.00	80.08–83.42
Respiratory rate	18.89 ± 1.68	13.00–26.00	18.59–19.20

**Table 2 jcm-11-07467-t002:** Intelligent rehabilitation assessment system score and physician’s FMA score.

Item		Physician’s FMA Score	Assessment System Score
Total score	Mean ± SD	41.48 ± 26.35	41.78 ± 25.42
	95% CI	36.72–46.25	37.19–46.38
Upper-limb score	Mean ± SD	24.03 ± 19.26	24.62 ± 18.35
	95% CI	20.55–27.51	21.27–27.97
Lower-limb score	Mean ± SD	17.45 ± 8.79	17.17 ± 9.21
	95% CI	15.86–19.04	15.50–18.83

**Table 3 jcm-11-07467-t003:** Regression analysis parameters of intelligent rehabilitation assessment system score and physician’s FMA score.

Item	Intercept	Slope	95% CI of Slope	R^2^	*p* Value
Total score	2.43	0.95	0.92–0.98	0.9667	0.0001 **
Upper-limb score	1.98	0.94	0.90–0.98	0.9577	0.0001 **
Lower-limb score	0.22	0.97	0.90–1.04	0.8575	0.0001 **

Total score: R^2^ = 0.9667, Y = 2.43 + 0.95X; upper-limb score: R^2^ = 0.9577, Y = 1.98 + 0.94X; lower-limb score: R^2^ = 0.8575, Y = 0.22 + 0.97X. Significant level at ** *p* < 0.001.

**Table 4 jcm-11-07467-t004:** The score deviation and relative deviation (%) of the intelligent rehabilitation assessment system and the physician’s FMA score.

Item		Score Deviation	Relative Deviation (%)
Total score	Mean ± SD	0.30 ± 4.83	3.49 ± 16.55
	95% CI	−0.57–1.17	0.50–6.48
Upper-limb score	Mean ± SD	0.58 ± 3.97	7.20 ± 26.77
	95% CI	−0.13–1.30	2.36–12.04
Lower-limb score	Mean ± SD	−0.28 ± 3.49	−0.78 ± 25.77
	95% CI	−0.91–0.35	−5.44–3.88

Score deviation = physician’s FMA score − intelligent rehabilitation assessment system score; relative deviation = (physician’s FMA score − intelligent rehabilitation assessment system score)/physician’s Fugl–Meyer score × 100%.

**Table 5 jcm-11-07467-t005:** Analysis of data distribution of deviation and relative deviation between intelligent rehabilitation assessment system score and physician’s Fugl–Meyer score.

Item		Deviation Distribution (%)	Relative Deviation Distribution (%)
Total score	Outside of Mean ± 1.96 SD	9 ± 7.50	5 ± 4.17
	Within Mean ± 1.96 SD	111 ± 92.50	115 ± 95.83
Upper score	Outside of Mean ± 1.96 SD	6 ± 5.00	5 ± 4.17
	Within Mean ± 1.96 SD	114 ± 95.00	115 ± 95.83
Lower score	Outside of Mean ± 1.96 SD	5 ± 4.17	4 ± 3.33
	Within Mean ± 1.96 SD	115 ± 95.83	116 ± 96.67

## Data Availability

The datasets used in this paper are all available on request from the author.

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
