# Peer review of "Wearable Intelligent Machine Learning Rehabilitation Assessment for Stroke Patients Compared with Clinician Assessment"

_jcm, 2022, doi:10.3390/jcm11247467_

Round 1

Reviewer 1 Report

This study aimed to explore the intelligent rehabilitation assessment system based on wearable devices and machine learning algorithms, and evaluate the effectiveness of the system in assessing stroke patients' rehabilitation outcomes. This topic could be also interesting for readers; however, several methodology flaws and unclear issues should be addressed by authors previously to be considered the manuscript for publication. The followings are few of my comments.  

1.      What were the specifications and manufacturers of wearable devices based on nine-axis inertial sensor Inertial Measurement Unit (IMU) and hand Flex sensor?  How to ensure good sensor quality data in this study?

2.      How the PC-side rehabilitation assessment software to estimate hemi-body 3D kinematics (e.g., each body segment orientation, relative segment position and joint angles)? How about the accuracy for position and for orientation? 

3.      Had there developed metrics for classifying activities and assessing the quality of lower and upper extremity movements were applied? It might be needed to present more in detail about how the signal processing and machine learning tools to quantitatively estimate the score for functional tasks from wearable IMU and hand Flex sensor data. Measurement reports are generated in an offline environment?

4.      The intelligent rehabilitation assessment system could significantly shorten the rehabilitation assessment time of patients, but from clinical practice point of view, it might also consider each system apply and perform verification time.

5.      Line 214 (1.2) [ y1,…..yn]

Author Response

1. What were the specifications and manufacturers of wearable devices based on nine-axis inertial sensor Inertial Measurement Unit (IMU) and hand Flex sensor?  How to ensure good sensor quality data in this study? Response 1:       Thank you for the reminder.       The manufacturer of wearable devices was Xi'an Libang Contmedu Medical Technology Co.,Ltd.,China, and the specification was KMD-T001.       The wearable devices were tested by third-party inspectors and met clinical research requirements in terms of signal quality, EMC and safety. In terms of measurement accuracy, according to the third-party inspection report, the measurement accuracy of angle, acceleration, and angular velocity are all above 95%.       It has been supplemented in the paper according to your suggestion.   2. How the PC-side rehabilitation assessment software to estimate hemi-body 3D kinematics (e.g., each body segment orientation, relative segment position and joint angles)? How about the accuracy for position and for orientation?   Response 2:       That is really a key issue. In the assessment of motor function, the joint angles and the acceleration and angular velocity of the limbs are the key information, which can be directly obtained or by simply calculation from the IMU or Flex sensor, with accuracy more than 95%. In addition, considering the time-consuming 3D reconstruction and kinematic analysis and the high requirements for computer even with GPU, 3D kinematic analysis was not performed in the rehabilitation assessment software.       Thank you for your advice. The relevant content has been supplemented in the paper.   3-1. Had there developed metrics for classifying activities and assessing the quality of lower and upper extremity movements were applied? Response 3-1:       The purpose of this study was to investigate the motor assessment function of stroke patients through the intelligent rehabilitation assessment system, and the study was not conducted for motor classification because the 32 movements were already set in order during the assessment process.       Considering that single movement score of the intelligent rehabilitation assessment system is 0-100 while the corresponding FMA score has only three values of 0,1,2, direct correspondence will lead to large error. Therefore, the scores of 32 movements were used as the input features of machine learning to obtain the total system score, and compared with the rehabilitation physician's FMA score to evaluate the designed intelligent rehabilitation assessment system.       Thank you for your advice. The relevant content has been supplemented in the paper.   3-2. It might be needed to present more in detail about how the signal processing and machine learning tools to quantitatively estimate the score for functional tasks from wearable IMU and hand Flex sensor data. Response 3-2:       Your suggestion makes the research process of this paper more rigorous.            Thank you very much. Relevant content has been added according to your suggestions.   3-3. Measurement reports are generated in an offline environment? Response 3-3:       Yes,measurement reports are generated in an offline environment, considering the application in the case without network.       Thanks for your suggestion, a description has been added to the paper.   4. The intelligent rehabilitation assessment system could significantly shorten the rehabilitation assessment time of patients, but from clinical practice point of view, it might also consider each system apply and perform verification time. Response 4:       Thanks a lot for your advice. Considering that the preparation time required for using the intelligent rehabilitation assessment system was about 1 minute, such as wearing devices and opening software, 1 minute preparation time had been included in the results and reports.       The relevant content has been supplemented in the paper.   5. Line 214 (1.2) [ y1,…..yn] Response 5:       Thank you very much for your careful review of this paper. It was our mistake and it has been modified in the paper.         We have completely revised the full text of this paper. For all revisions to the manuscript, the revision markers are used so that you can see the changes more clearly.       Thank you again for your positive comments and constructive guidance on this paper, and hope to receive your positive response.    

Reviewer 2 Report

In this paper, a new ‘wearable device’ rehabilitation  assessment system was used with 120 stroke patients, and related to the Fugl-Meyer scores obtained from assessment of the same patients by rehabilitation clinicians. A high level of agreement was obtained between the two methods.

That is, the paper is a validation study of a new instrumented stroke rehabilitation assessment tool. Good reasons for the logic and motivation underlying the development of the assessment system and software are put forward and the positive results are well-summarized in sections 6 and 7 (although the results here come from 120 clinical ‘applications’, not ‘clinical trials’ – Line 433).

What the research is not, is a clinical trial, with an Experimental Group and a Control Group that participants are randomized into (e.g. getting or not getting a drug or a treatment), even though this is what the text says on Line 286.

The text goes on to note that there were no significant differences between the Experimental Group and Control Group on various parameters (Line 302). Apart from the headings in Table 1 (Line 308), these two groups are never mentioned again. It would appear that the two groups were in fact the two groups of 60 stroke patients each, from Tangdu Hospital and Xi’an Gaoxin Hospital. While there is benefit to generalization of results if similar findings are obtained from two different sites, the two hospital groups are never compared. There is also benefit to generalizability if similar findings are obtained with different teams in different hospitals, however, it would seem that the same team of application technicians and rehabilitation clinicians did all assessments, with the clinicians standing to one side and observing when the intelligent system assessment was being conducted. This is fine for an assessment validation study.

My problem, therefore, is that a validation study comparing two stroke-patient assessment systems, one device-dependent and one clinician-dependent, is explicitly confused with a randomized clinical trial, in the title, and in first half of the paper.

I would suggest, therefore, that the authors completely rewrite the paper, with the title….

 ‘Wearable intelligent machine-learning rehabilitation assessment for stroke patients compared with clinician assessment’.

All reference to clinical trials with Experimental and Control Groups should be removed.

Author Response

1.My problem, therefore, is that a validation study comparing two stroke-patient assessment systems, one device-dependent and one clinician-dependent, is explicitly confused with a randomized clinical trial, in the title, and in first half of the paper.

Response 1:

Thanks a lot for your advice, which solved my previous doubts.The title of the paper and the content related to clinical trials in the first half of the paper have been revised according to your suggestions.   2.I would suggest, therefore, that the authors completely rewrite the paper, with the title…. ‘Wearable intelligent machine-learning rehabilitation assessment for stroke patients compared with clinician assessment’. Response 2: Thank you very much for your advice. The title has been revised according to your suggestion.   3.All reference to clinical trials with Experimental and Control Groups should be removed. Response 3: Thank you for your advice. The relevant content has been revised according to your suggestions.   We have completely revised the full text of this paper. For all revisions to the manuscript, the revision markers are used so that you can see the changes more clearly.   Thank you again for your positive comments and constructive guidance on this paper, and hope to receive your positive response.

Round 2

Reviewer 2 Report

Thank you for quickly and effectively making revisions to this paper.

I think that it is a valuable addition to the literature, and will generate discussion and further research.